# Effect of Cancer-Related Cachexia and Associated Changes in Nutritional Status, Inflammatory Status, and Muscle Mass on Immunotherapy Efficacy and Survival in Patients with Advanced Non-Small Cell Lung Cancer

**DOI:** 10.3390/cancers15041076

**Published:** 2023-02-08

**Authors:** Clelia Madeddu, Silvia Busquets, Clelia Donisi, Eleonora Lai, Andrea Pretta, Francisco Javier López-Soriano, Josep Maria Argilés, Mario Scartozzi, Antonio Macciò

**Affiliations:** 1Medical Oncology Unit, “Azienda Ospedaliero Universitaria” of Cagliari, Department of Medical Sciences and Public Health, University of Cagliari, 09100 Cagliari, Italy; 2Departament de Bioquímica i Biomedicina Molecular, Facultat de Biologia, University of Barcelona, Diagonal 643, 08028 Barcelona, Spain; 3Institut de Biomedicina de la Universitat de Barcelona (IBUB), 08028 Barcelona, Spain; 4Gynecologic Oncology Unit, ARNAS G. Brotzu, Department of Surgical Sciences, University of Cagliari, 09100 Cagliari, Italy

**Keywords:** immune checkpoint inhibitor, immunotherapy, non-small cell lung cancer, survival, inflammation, cachexia, sarcopenia, IL-6, glasgow prognostic score, NLR, miniCASCO

## Abstract

**Simple Summary:**

Immune checkpoint inhibitor (ICI)-based immunotherapy has dramatically improved the survival of patients with advanced non-small cell lung cancer (NSCLC); however, a significant percentage of these patients do not benefit from this approach. Therefore, predictive biomarkers are needed. Increasing evidence demonstrates that cancer cachexia, which is often reported in patients with NSCLC, with associated chronic inflammation and related changes in body composition, metabolism, and nutritional status, may affect the immune response and impair immunotherapy efficacy. However, few prospective studies have explored the association between cachexia and the response to immunotherapy in NSCLC. We designed a prospective observational study to evaluate the prognostic and predictive role of cachexia, with its related changes in inflammatory, immunological, and nutritional parameters, on the survival and clinical response (i.e., disease control rate) to ICI in patients with advanced NSCLC. Our results suggest that cachexia can be an independent unfavorable prognostic and predictive factor. Then, the evaluation of cachexia should be strongly considered as a key parameter in the design of immunotherapy-based trials.

**Abstract:**

Immune checkpoint inhibitor (ICI)-based immunotherapy has significantly improved the survival of patients with advanced non-small cell lung cancer (NSCLC); however, a significant percentage of patients do not benefit from this approach, and predictive biomarkers are needed. Increasing evidence demonstrates that cachexia, a complex syndrome driven by cancer-related chronic inflammation often encountered in patients with NSCLC, may impair the immune response and ICI efficacy. Herein, we carried out a prospective study aimed at evaluating the prognostic and predictive role of cachexia with the related changes in nutritional, metabolic, and inflammatory parameters (assessed by the multidimensional miniCASCO tool) on the survival and clinical response (i.e., disease control rate) to ICI-based immunotherapy in patients with advanced NSCLC. We included 74 consecutive patients. Upon multivariate regression analysis, we found a negative association between IL-6 levels (odds ratio (OR) = 0.9036; 95%CI = 0.8408–0.9711; *p* = 0.0025) and the miniCASCO score (OR = 0.9768; 95%CI = 0.9102–0.9999; *p* = 0.0310) with the clinical response. As for survival outcomes, multivariate COX regression analysis found that IL-6 levels and miniCASCO-based cachexia severity significantly affected PFS (hazard ratio (HR) = 1.0388; 95%CI = 1.0230–1.0548; *p* < 0.001 and HR = 1.2587; 95%CI = 1.0850–1.4602; *p* = 0.0024, respectively) and OS (HR = 1.0404; 95%CI = 1.0221–1.0589; *p* < 0.0001 and HR = 2.3834; 95%CI = 1.1504–4.9378; *p* = 0.0194, respectively). A comparison of the survival curves by Kaplan–Meier analysis showed a significantly lower OS in patients with cachexia versus those without cachexia (*p* = 0.0323), as well as higher miniCASCO-based cachexia severity (*p* = 0.0428), an mGPS of 2 versus those with a lower mGPS (*p* = 0.0074), and higher IL-6 levels (>6 ng/mL) versus those with lower IL-6 levels (≤6 ng/mL) (*p* = 0.0120). In conclusion, our study supports the evidence that cachexia, with its related changes in inflammatory, body composition, and nutritional parameters, is a key prognostic and predictive factor for ICIs. Further larger studies are needed to confirm these findings and to explore the potential benefit of counteracting cachexia to improve immunotherapy efficacy.

## 1. Introduction

Lung cancer is the leading cause of cancer-related deaths worldwide [1]. The use of immunotherapy, particularly immune checkpoint inhibitors (ICIs), such as anti-programmed cell death protein-1 (anti-PD-1) and anti-PD1 ligand (anti-PD-L1) antibodies, has been approved for both first- and second-line treatments of advanced (stage IIIB/IV) non-small cell lung cancer (NSCLC) with relevant improvement in patient survival. Nonetheless, only 10%–20% of patients with advanced NSCLC demonstrate durable treatment responses [2,3]. Therefore, predictive biomarkers of the response to treatment are needed for the prompt and efficient identification of patients who are more likely to benefit from immunotherapy. Recent studies have provided increasing evidence demonstrating that a patient’s general condition-related factors, including body composition and nutritional/inflammatory status, may affect the immune response efficiency and immunotherapy efficacy [4,5,6,7,8,9,10]. Different authors have evaluated the effect of body composition changes, particularly sarcopenia, on the chemotherapy and immunotherapy response and prognosis in patients with different types of solid cancers, especially NSCLC [11,12,13,14,15,16]. Sarcopenia is a key component of cancer-related anorexia cachexia syndrome (CACS) in patients with advanced cancer. CACS, a complex multifactorial syndrome associated with and sustained by chronic inflammation, is characterized by the dysfunctional use of nutrients, along with an increased and altered energy metabolism, thereby leading to involuntary weight loss with an ongoing decrease in skeletal muscle mass and depletion of body resources [17]. The main clinical feature of CACS is weight loss with lean mass reduction (sarcopenia), which is usually associated with other signs and symptoms such as anorexia, fatigue, and anemia [18]. Consistently, the definition of cachexia and its severity should include several domains: anorexia or reduced food intake, catabolic drive (inflammation and tumor growth), loss of muscle mass and muscle strength, and functional or psychosocial impairment. Moreover, when evaluating the prognostic and predictive role of CACS, it is important to consider that this syndrome develops progressively through various stages, from pre-cachexia to cachexia and refractory cachexia [18]. Among the different tools available to classify CACS, the CAchexia SCOre (CASCO) evaluates cachexia through a set of variables (anthropometric and body composition-related variables; inflammatory, immunosuppressive, and metabolic variables; nutritional and dietary variables; functional and physical activity-related variables; and quality of life [QoL]) staging for cachexia severity. The CASCO was published in 2011; subsequently, a shortened and more feasible version of the score, the miniCASCO, was developed and validated [19].

CACS occurs in over 80% of patients with advanced cancer and is responsible for approximately 20% of cancer-related deaths. In patients with lung cancer, the prevalence of sarcopenia and cachexia ranges from 40% to 80%, which is higher than the prevalence of sarcopenia and cachexia in patients with most other cancer types [20,21,22].

Cancer-related inflammation is recognized as the main pathogenetic [23] and key defining factor of CACS [24], being responsible for altering the peripheral energy metabolism through various pathways, inducing insulin resistance, enhancing lipolysis, and protein catabolism, and promoting the loss of lean muscle mass and anorexia [17]. In turn, cancer-related inflammation with related metabolic and nutritional changes can contribute to immunosuppression by different mechanisms [25,26,27] and can thus considerably affect the antineoplastic immune response and PD-1/PD-L1 inhibitor efficacy, involving poorer outcomes in patients with cancer cachexia treated with these agents.

Hence, baseline cachexia with associated changes in body composition (weight loss and sarcopenia) has been hypothesized to be a negative predictor of the response to ICI treatment. Hitherto, some retrospective studies have examined the role of sarcopenia alone in the immunotherapy response in patients with NSCLC [28,29,30,31,32,33,34]. However, few small-scale prospective studies have investigated the effect of cachexia syndrome, with its multidimensional components, on immunotherapy efficacy in patients with lung cancer. Indeed, evaluating the specific effect of CACS on immunotherapy efficacy requires the concomitant appraisal of several variables (associated with cachexia and its severity) that can affect the ICI treatment response.

Therefore, we designed a prospective observational study aimed at evaluating the prognostic or predictive role of cachexia (with related changes in the inflammatory, immunological, and nutritional parameters) on the survival (progression-free survival [PFS] and overall survival [OS]) and clinical response (i.e., disease control rate) to anti-PD-1 inhibitor-based monotherapy in patients with advanced NSCLC. We used the miniCASCO as a multidimensional tool to stage and classify cachexia severity.

## 2. Materials and Methods

### 2.1. Study Design

This was a prospective, observational study of patients with advanced NSCLC treated with anti-PD-1/PD-L1 monotherapy in a clinical setting, regardless of treatment line. The patients were treated with the following standard doses and schedules of pembrolizumab or nivolumab: 200 mg of pembrolizumab every three weeks and 240 mg of nivolumab every two weeks.

From March 2017 to August 2021, we consecutively screened all patients with metastatic NSCLC in the Oncology Unit of the University Hospital in Cagliari (Azienda Ospedaliero Universitaria di Cagliari, Cagliari, Italy), who were eligible to receive monotherapy with nivolumab or pembrolizumab. For study enrolment, patients were required to have measurable disease per the Response Evaluation Criteria in Solid Tumors (RECIST) version 1.1. The objective clinical response to immunotherapy was classified based on the RECIST version1.1. The first objective clinical response evaluation took place during the eighth week of treatment. Subsequent tumor measurements were performed every eight weeks. Moreover, we calculated the disease control rate (DCR) and the overall response rate (ORR). DCR was defined as the sum of the proportion of patients with a complete response (CR), partial response (PR), and stable disease (SD) at the first tumor measurement (week eight). The ORR was defined as the sum of the proportion of complete and partial responder patients at the first tumor measurement (week eight). PFS was calculated from the first day of immunotherapy administration until the progression of disease (PD) or death due to any cause. OS was defined as the time elapsed between evaluation and death or the last visit. Indeed, patients were followed until death. For PFS analysis, those patients alive without PD at the time of analysis were censored at the last visit. For OS analysis, those patients alive at the time of analysis were censored at the last visit. Patients who experienced PD after ICIs treatment underwent further standard approved antineoplastic treatments or best supportive care based on their clinician’s evaluation and choice: the OS data of these patients were included in the analysis. For each tumor type, we performed PD-L1 immunostaining on freshly cut slides from representative blocks using an anti-PD-L1 antibody, Ventana SP263 Assay (Roche Diagnostics, Penzberg, Germany). The percentages of membranous PD-L1-positive tumor cells were evaluated for each sample.

All patients provided written informed consent for immunotherapy and study participation. Moreover, the study was conducted following the guidelines for Good Clinical Practice and the principles of the Declaration of Helsinki; study approval was provided by the Local Institutional Ethics Committee Azienda Ospedaliero Universitaria di Cagliari.

### 2.2. Eligibility Criteria

We included patients that met the following criteria: Stage IV histologically proven NSCLC eligible for nivolumab or pembrolizumab monotherapy, age ≥18 years, measurable disease according to RECIST 1.1 assessed by CT before starting the immunotherapy (no more than one month earlier), ECOG PS 0–2, and laboratory liver and renal function values in accordance with standardized approved criteria for ICI treatment (bilirubin, alkaline phosphatase and transaminase levels < 1.5 × normal upper limits; sodium > 125 mmol/L; normal calcium; creatinine clearance > 40 mL/min). According to drug-approved indications, the patients were eligible either for first-line pembrolizumab if having a PD-L1 score ≥ 50% of tumor cells or for a subsequent line after having experienced a failure of first-line platinum-based chemotherapy; these latter patients received monotherapy with pembrolizumab (pending on PD-L1 score > 1% of tumor cells) or nivolumab (whatever the PD-L1 score). In addition, the exclusion criteria were as follows: active malignancy other than NSCLC, EGFR/ALK/ROS1 oncogene-addicted NSCLC, diagnosis of concomitant autoimmune disease in an active phase, previous or concomitant episode of thyroiditis or hypophysitis, acute cardiac failure and unstable coronary angina, presence of symptomatic brain metastases or metastases requiring high-dose steroid therapy, serological positivity for hepatitis B or C viruses and HIV, baseline aspartate amino transferase levels >2.5 times the normal levels and baseline total bilirubin levels ≥3 times the normal levels, pregnant women or lactating mothers, and inability to provide verbal or written informed consent.

### 2.3. Outcomes

The primary outcomes were the correlations of cachexia, miniCASCO score, miniCASCO-based cachexia severity, nutritional status, inflammatory status, SMI with the DCR, PFS, and OS. The secondary outcome was a comparison of survival outcomes (PFS and OS) between groups based on GPS, sarcopenia status, and miniCASCO grading of cachexia severity. As a corollary outcome, we evaluated the association between the RECIST objective clinical response category at the first CT evaluation (week eight) and changes in the body composition and nutritional and laboratory parameters assessed.

### 2.4. Collection of Clinical and Laboratory Data

All of the patients were evaluated at baseline before treatment initiation. The clinical data collected were anthropometric parameters (age, sex, weight, height, and BMI), tumor histology and stage, PD-L1 status, and Eastern Cooperative Oncology Group performance status (ECOG PS). The baseline stage and extent of measurable disease were evaluated by total body CT performed before starting immunotherapy (no more than one month earlier). We also evaluated laboratory parameters related to inflammation and nutritional status (hemoglobin level, absolute neutrophil count, absolute lymphocyte count, NLR, CRP level, and serum albumin level), modified Glasgow prognostic score (mGPS) (mGPS = 2, both elevated CRP [≥10 mg/L] and low serum albumin levels [<3.5 g/dL]; mGPS = 1, elevated CRP levels only; mGPS = 0, normal CRP levels [<10 mg/L]), lean body mass (LBM), skeletal muscle mass index (SMI), weight loss, cachexia defined using the Fearon criteria [18], and miniCASCO-evaluated cachexia severity.

According to Fearon et al. [18], the diagnostic criteria for cachexia are ≥5% weight loss or ≥2% weight loss in individuals already showing decreases in body weight and height (body mass index [BMI] < 20 kg/m2) or skeletal muscle mass (sarcopenia).

The estimation of LBM and SMI was performed according to the methods detailed below on both the CT performed before starting immunotherapy and on the first CT evaluation of the objective clinical response performed at week eight after treatment started. Peripheral blood samples were collected from the included patients just before the first immunotherapy infusion. Data were assessed at baseline (T0) and at the time of the first instrumental evaluation of the objective clinical response at week eight (T1), and the delta value, i.e., the difference between the T1 and T0 values, was calculated. The blood analyses were performed at the Central Laboratory of the University Hospital, according to standard methods, and subject to periodic controls. The coefficients of variation for these methods, following quality control procedures, were less than 5%.

#### 2.4.1. Assessment of Skeletal Muscle Mass and SMI

The cross-sectional areas of lumbar skeletal muscles at the third lumbar vertebra (L3) level were analyzed using electronically stored CT images. LBM and SMI were analyzed using slice-O-matic software version 5.0 (Tomovision, Montreal, Canada) and calculated based on the equations provided by Mourtzakis et al. [35] as reported in detail in the Appendix A.

The SMI cut-off values used for the definition of sarcopenia were set at <55 cm^2^/m^2^ and <39 cm^2^/m^2^ for men and women, respectively, according to the international consensus for the definition of cancer cachexia [18].

#### 2.4.2. CT image Analysis

For the SMI calculation, a single-slice CT image at L3, with both transverse processes visible, was selected. Specific tissue demarcation was performed on the image using the following Hounsfield unit (HU) thresholds: −29 to +150 (skeletal muscle excluding visceral organs), −190 to −30 (subcutaneous and intramuscular adipose tissue), and −150 to −50 (visceral adipose tissue). The TMA and total fat tissue area (subcutaneous and visceral adipose tissues) were assessed.

#### 2.4.3. Assessment of Cachexia Using the miniCASCO

We assessed the cachexia severity using the miniCASCO questionnaire and its subscales: bodyweight composition/change (BWC), inflammation/metabolic disturbances/immunosuppression (IMD), physical performance (PHP), anorexia, and QoL. The miniCASCO was used to classify patients into four categories: no (≤14), mild (15–28), moderate (29–46), and severe (>46) cachexia, respectively [19]. The patients reported symptoms, treatment side effects, and reductions in their activities of daily living using the miniCASCO questionnaire. The miniCASCO-related data were analyzed using an online software (https://www.ub.edu/cancerresearchgroup/casco.php, accessed on 1 December 2022; http://hdl.handle.net/2445/65137, accessed on 1 December 2022). The total score of miniCASCO is the sum of each single subscale, where each subscale contributes to the final score with a different weight, as indicated in detail in the validation paper [19].

### 2.5. Statistical Analysis

Continuous variables were assessed for linearity through the Kolmogorov–Smirnov test. Linear variables are reported as means and standard deviations, while non-linear variables are reported as median and range. Categorical variables were reported as absolute numbers and percentages. The differences between groups were assessed using the Student’s *t*-test and chi-squared or Fisher’s exact test. Differences between multiple groups were assessed by ANOVA or Kruskal–Wallis’ test for linear or non-linear variables, respectively. Correlation between continuous variables was assessed by Pearson or Spearman correlation analysis for parametric or not parametric data, respectively. Probabilities and odds ratios (ORs) of a change in categorically dependent variables, namely, disease control versus progressive disease at the eighth week, conditional on the values of independent covariables, were analyzed in a logistic regression model. Variables associated with a *p*-value of <0.5 in the univariate analyses were included in the multivariate models, i.e., multivariate regression analysis (stepwise method), considering the clinical response as the dependent variable. Cox proportional hazards regression analysis was used to identify those clinical and biological variables predictive of PFS or OS. Possible factors identified by univariate regression analysis were evaluated using multivariate Cox regression analysis and the stepwise method to determine the independent predictors of PFS and OS rates. Survival curves were compared using the Kaplan–Meier (log-rank) test. The cut-off of continuous variables predictive of PFS and OS, established to compare survival curves by the Kaplan–Meier method, was the median value. 

A two-tailed *p*-value of <0.05 was considered statistically significant. All statistical analyses were performed using MedCalc Statistical Software version 20.115 (2022 MedCalc Software Ltd., Ostend, Belgium).

## 3. Results

### 3.1. Patient Characteristics

A total of 134 patients with advanced (stage IV) non-oncogene-addicted NSCLC, candidates to receive immunotherapy with nivolumab or pembrolizumab monotherapy, were identified and screened between March 2017 and August 2021. A total of 60 patients were excluded from the final analysis: 24 patients lacked information regarding body weight; 20 patients were excluded for ECOG PS>2 in accordance with regulator pharmaceutical agency approval for treatment with nivolumab and pembrolizumab; 2 patients were enrolled in a clinical trial; 5 patients lacked the lumbar CT images at L3 level before treatment; 5 patients did not have the CT evaluation at week eight after treatment; and 4 patients were transferred to another hospital during treatment. Finally, we included in the study analysis 74 patients (Appendix A). The clinical characteristics of the enrolled patients at diagnosis are presented in Table 1. The mean patient age was 69.3 ±11.3 years; at the time of evaluation, 57 (77%) patients had histologically proven adenocarcinoma, and 22 patients (29.7%) had brain metastases. In all, 32 (43.2%), 22 (29.8%), 10 (13.5%), and 10 (13.5%) patients had tumor PD-L1 expression of >50%, tumor PD-L1 expression between 1% and 50%, tumor PD-L1 expression of <1%, and unknown PD-L1 status, respectively. Forty-nine (66%) patients had an ECOG PS of 0 or 1; twenty-five (34%) patients had an ECOG PS of 2. No patient had an ECOG PS of 3 or 4 because the criteria for outpatient treatment with ICIs (nivolumab or pembrolizumab) require PS ≤ 2 in accordance with regulatory pharmaceutical agency indication for these specific drugs. Thirty-two (43%) patients underwent one previous line of treatment with standard first-line chemotherapy with platinum-based regimens (e.g., carboplatin/paclitaxel, carboplatin/gemcitabine, or carboplatin/pemetrexed). No patients underwent previous surgery and/or radiotherapy.

During the follow-up period (median 24 months, range 5–63+), 50 death events (67% of patients) were observed; the median PFS was 15 months (mean 20.5 months; range, 2–63+ months), and the median OS was 24 months (mean 20.6 months; range, 5–63+ months). Twenty patients had ongoing objective responses at data cut-off and thus continued to receive treatment (ORR = 35.1%; DCR = 71.6%). The best responses were PD, SD, PR, and CR in 21 (28.4%), 27 (36.5%), 22 (29.7%), and 4 (5.4%) patients, respectively.

#### 3.1.1. Distribution of Patients Based on BMI and the Presence of Weight Loss and Sarcopenia

In all, 4 (5.4%), 38 (51.4%), 24 (32.4%), and 8 (10.8%) patients were underweight (BMI <18.5 kg/m^2^), normal weight (18.5 ≤ BMI < 25 kg/m^2^), overweight (25 ≤ BMI < 30 kg/m^2^), and obese (BMI ≥ 30 kg/m^2^), respectively. The mean percentage of weight loss at baseline was 4.2% (range, 0–16%). Thirty-nine (52.9%) patients satisfied the cachexia criteria proposed by Fearon et al. [18]. Notably, 51 of the 74 (68.9%) enrolled patients had sarcopenia; moreover, all individuals with obesity had sarcopenia (Table 2). Of note, we did not observe a significant difference in the incidence of cachexia (according to the Fearon criteria) at enrolment between those patients who received one previous line of treatment and those who did not (*p* = 0.3879).

#### 3.1.2. Baseline Body Composition and Nutritional and Inflammatory Parameters

Table 2 presents the body composition (LBM and SMI) and nutritional and inflammatory status data. A high proportion of patients had an mGPS of 2, indicating the presence of an inflammatory status associated with a compromised nutritional status (20 (27%), 20 (27%), and 34 (46%) patients had an mGPS of 0, 1, and 2, respectively).

**Table 2 cancers-15-01076-t002:** Parameters of body composition and nutritional/inflammatory status at baseline.

Parameter (Normal Value)	Mean Value ± SD (Range)	N. (%)
BMI, kg/m^2^		
<18.5		4 (5.4)
18.5–25		38 (51.4)
≥25–30		24 (32.4)
≥30		8 (10.8)
LBM, kg (NA)	30.5 ± 9.3 (15–48.6)	
SMI, cm^2^/m^2^	43.9 ± 12.8 (24–69)	
Sarcopenia		51 (68.9)
SMI ≤39 cm^2^/m^2^ for women		
SMI ≤55 cm^2^/m^2^ for men		
CRP, mg/L (0–5)	22.5 ± 16.7 (0.1–150.8)	
IL-6, pg/mL (0–7)	12.8 ± 3.3 (1–85)	
Neutrophils, cells/microl (5000–10000)	5700 ± 2578.7 (2000–12800)	
Lymphocytes, cells/microl (1200–4000)	1703 ± 580.9 (600–3300)	
NLR (NA)	3.8 ± 2.3 (1–12)	
Albumin, g/dL (>3.2)	3.5 ± 0.4 (2.6–4.4)	
Hb, g/dL (≥12)	12.2 ± 1.9 (7.7–16.3)	
mGPS		
0		20 (27)
1		20 (27)
2		34 (46)

Data are reported as the mean ± standard deviation (SD) for continuous variables or number and percentage for categorical variables. Abbreviations: BMI, body mass index; SMI, skeletal mass index; CRP, C-reactive protein; IL, interleukin; NLR, neutrophil-to-lymphocyte ratio; Hb, hemoglobin; mGPS, modified Glasgow Prognostic Score.

#### 3.1.3. Analysis of the Findings of the miniCASCO Questionnaire and Its Subscales

The mean miniCASCO at baseline was 32.5 ± 16.4 (range, 6–73). The mean subscale scores are presented in Appendix A. The evaluation of the miniCASCO-based cachexia severity showed no, mild, moderate, and severe cachexia in 13 (17.6%), 19 (25.7%), 30 (40.5%), and 12 (16.2%) patients, respectively. Of note, we did not observe a significant difference in the miniCASCO score at enrolment between those patients who received one previous line of treatment and those who did not (*p* = 0.998).

### 3.2. Association between Classic Prognostic Factors (PD-L1 Expression and ECOG PS) and Clinical Outcomes (Objective Clinical Response, PFS, and OS)

Upon logistic regression analysis, no significant association was found between PD-L1 expression and the objective clinical response categories, i.e., disease control versus progressive disease (regression coefficient = 0.12033; OR = 1.1275; 95%CI: 0.5652 to 2.2492; *p* = 0.7333) nor between the ECOG PS and objective clinical response categories, i.e., disease control versus progressive disease (regression coefficient = −0.35610; OR = 0.7004; 95%CI = 0.2883–1.7018; *p* = 0.4318).

COX regression analysis did not show a significant association between PD-L1 expression and PFS (b coefficient = −0.09548; Exp(b) = 0.9089; 95%CI = 0.6109–1.3524; *p* = 0.6377) or OS (b coefficient = 0.1565; Exp(b) = 1.1694; 95%CI = 0.7441–1.8377; *p* = 0.4975), nor between ECOG PS and PFS (b coefficient = 0.4670; Exp(b) = 1.5952; 95%CI = 0.9982–2.5492; *p* = 0.0509) or OS (b coefficient = 0.4235; Exp(b) = 1.5273; 95%CI = 0.9032–2.5828; *p* = 0.1141). Consistently, the ANOVA results did not show a significant association between PD-L1 expression and PFS (*p* = 0.388336) or OS (*p* = 0.563), nor between ECOG PS and PFS (*p* = 0.121) or OS (*p* = 0.303). The lack of association between PS and clinical outcomes could have been influenced by the inclusion of only patients with ECOG PS 0-2 as requested by the established regulatory criteria for eligibility for anti-PD1 therapy with nivolumab and pembrolizumab.

### 3.3. Association between Cachexia Status, miniCASCO, Body Composition, Inflammatory Status, and Nutritional Status and the Clinical Response (Disease Control Rate)

The univariate logistic regression analysis found an inverse association between the objective clinical response categories (i.e., disease control versus progressive disease) and the miniCASCO score (regression coefficient=−0.1651; OR = 0.9035; 95%CI = 0.6766–0.9845; *p* = 0.0186), including the IMD subscale score (regression coefficient = −0.0466; OR = 0.9545; 95%CI = 0.8565–0.9988; *p* = 0.0469), as well as CRP (regression coefficient = −0.672; OR = 0.9971; 95%CI = 0.8164–0.9848; *p* = 0.0227), IL-6 (regression coefficient = −0.0799; OR = 0.9232; 95%CI = 0.8762–0.9728; *p* = 0.0027), and neutrophil levels (regression coefficient = −0.00021805; OR = 0.9998; 95%CI = 0.9996–1.0000; *p* = 0.0403) (Table 3). A positive significant association was found between the ANO subscale and clinical response categories, i.e., disease control versus progressive disease (regression coefficient = 0.1607; OR = 1.1743; 95%CI = 1.0056–1.3713; p = 0.0423). Multivariate regression analysis showed that both the IL-6 levels (OR = 0.9036; 95%CI = 0.8408–0.9711; *p* = 0.0025) and miniCASCO (OR = 0.9768; 95%CI = 0.9102–0.9999; *p* = 0.0310) were independent predictors of the clinical response.

### 3.4. Cox Regression Analysis of Variables Determinant of PFS and OS

The results of the univariate and multivariate COX regression analyses of OS and PFS are shown in Appendix A. In the univariate COX regression analysis, the IL-6 levels (b coefficient = 0.03155; Exp(b) = 1.0321; 95%CI = 1.0176–1.0468; *p* < 0.0001), mGPS (b coefficient = 0.4079; Exp(b) = 1.5037; 95%CI = 1.1087–2.0396; *p* = 0.0087), miniCASCO score (b coefficient = 0.07253; Exp(b) = 1.0752; 95%CI = 1.0137–1.1405; *p* = 0.0159), IMD subscale (b coefficient = 0.09010; Exp(b) = 1.0943; 95%CI = 1.0277–1.1652; *p* = 0.0049), QoL subscale (b coefficient = 0.1862; Exp(b) = 1.2046; 95%CI = 1.0516–1.3799; *p* = 0.0072), and miniCASCO-based cachexia severity (b coefficient = 0.2510; Exp(b) = 1.2854; 95%CI = 1.1244–1.7075; *p* = 0.0083) were found to be associated with PFS (Appendix A). Upon multivariate COX regression analysis (stepwise method), the IL-6 levels (b coefficient = −0.0386; adjusted Exp(b) = 1.0388; 95%CI = 1.0230–1.0548; *p* < 0.0001) and miniCASCO-based cachexia severity (b coefficient = 0.2301; adjusted Exp(b) = 1.2587; 95%CI = 1.0850–1.4602; *p* = 0.0024) significantly affected PFS (Table 4).

As regards OS, in the univariate analysis, cachexia according to the Fearon criteria (b coefficient = 0.1306; Exp(b) = 1.1396; 95%CI = 1.4632–1.8875; *p* = 0.0306), IL-6 levels (b coefficient = 0.03825; Exp(b) = 1.0390; 95%CI = 1.0213–1.0570; *p* < 0.0001), mGPS (b coefficient = −0.4847; Exp(b) = 1.6237; 95%CI = 1.1771–2.2396; *p* = 0.0031), miniCASCO score (b coefficient = −0.1619; Exp(b) = 1.1630; 95%CI = 1.0341–1.9989; *p* = 0.0047), IMD subscale (beta coefficient = −0.1021; Exp(b) = 1.1075; 95%CI = 1.0300–1.1908; *p* = 0.0058), and miniCASCO-based cachexia severity (b coefficient = 0.6530; Exp(b) = 1.9233; 95%CI = 1.0252–3.6080; *p* = 0.0416) were found to be associated with OS (Appendix A). Upon multivariate COX regression analysis (stepwise method), IL-6 (b coefficient = 0.03956; adjusted Exp(b) = 1.0404; 95%CI = 1.0221–1.0589; *p* < 0.0001) and miniCASCO-based cachexia severity (b coefficient = 0.8685; adjusted Exp(b) = 2.3834; 95%CI = 1.1504–4.9378; *p* = 0.0194) significantly affected OS (Table 4).

**Table 4 cancers-15-01076-t004:** Multivariate COX regression analysis of the variables associated with PFS and OS.

Variable	PFS		OS	
	Adjusted Exp(b) (95%CI)	*p-*Value	Adjusted Exp(b) (95%CI)	*p-*Value
IL-6	1.0388 (1.0230–1.0548)	<0.0001	1.0404 (1.0221–1.0589)	**<0.0001**
miniCASCO-based cachexia severity	1.2587 (1.0850–1.4602)	0.0024	2.3834 (1.1504–4.9378)	**0.0194**
Cachexia according to the Fearon criteria	-	-	0.7774 (0.4107–1.4715)	0.4392
mGPS	1.3171 (0.9881–1.5936	0.0056	1.3494 (0.8355–2.1796)	0.2205
miniCASCO	0.9797 (0.9428–1.0180)	0.2948	0.9884 (0.9479–1.0307)	0.5855
IMD subscale	0.9760 (0.8908–1.0693)	0.6018	1.0003 (0.9089–1.1009)	0.9955
QoL subscale	1.4822 (0.7517–2.9226)	0.2560	-	-

Significant *p*-values (<0.05) are reported in bold. Abbreviations: IL, interleukin; mGPS, modified Glasgow Prognostic Score; IMD, inflammation/metabolic disturbances/immunosuppression; QoL, quality of life.

Analysis of the association between cachexia status, miniCASCO score, and the parameters of body composition, inflammatory status, and nutritional status and the survival outcomes via correlation analysis for the continuous variables and via ANOVA (or Kruskall–Wallis) test for the categorical variables for the comparison of PFS/OS between categories, confirmed the results of the COX regression analysis (Appendix A, Appendix A, and Appendix A).

### 3.5. Kaplan–Meier Survival Analysis of OS and PFS in Terms of the Presence of Cachexia, miniCASCO-Based Cachexia Severity, Sarcopenia Status, GPS Categories, and IL-6 Values

Based on the presence of cachexia, patients with cachexia showed a significantly shorter OS (26.3 ± 3.4 months) than those without cachexia (40.2 ± 4.3 months) (unadjusted HR = 1.9665; 95%CI = 1.0586–3.6530; *p* = 0.0323) (Figure 1a).

The analysis of OS in terms of miniCASCO-based cachexia severity showed that patients without cachexia had a significantly longer OS (45.9 ± 6.9 months) than those with mild (OS = 30.8 ± 3.1 months; unadjusted HR = 0.3720; 95%CI = 0.1572–0.8800), moderate (25.9 ± 3.1 months; unadjusted HR = 0.3848; 95%CI = 0.1720–0.8612), and severe (26 ± 4.5 months; HR = 0.3914; 95%CI = 01,473–0.9401) cachexia (*p* = 0.0428) (Figure 1b).

The OS analysis based on the evidence of sarcopenia did not show a significant difference between patients with no sarcopenia in comparison to those with sarcopenia (34.7 ± 3.1 months versus 22.9 ± 3.7 months; unadjusted HR = 0.4508; 95%CI = 0.2125–1.0024; *p* = 0.0691).

The OS analysis based on mGPS showed that patients with an mGPS of 2 had a significantly lower mean OS (21.6 ± 2.9 months) in comparison to those with an mGPS of 0 (38.9 ± 3.9 months) and mGPS of 1 (21.7 ± 3.3 months) (unadjusted HR = 2.4539; 95%CI = 1.2866–4.6801; *p* = 0.0074) (Figure 1c). The OS analysis between patients divided on the basis of the median IL-6 value (i.e., 6 ng/mL) showed that those with an IL-6 > 6 ng/mL had a significantly lower PFS in comparison to those with an IL-6 ≤6 ng/mL (20.1 ± 2.7 versus 38.6 ± 3.7 months; unadjusted HR = 2.7680; 95%CI = 1.2504–6.1275; *p* = 0.0120) (Figure 1d).

As regards PFS, we did not find a statistically significant difference between patients categorized according to the presence of cachexia or not (26.4 ± 4.1 months versus 20.1 ± 3.4 months; HR = 0.6891; 95%CI = 0.3888–1.2213; *p* = 0.2021) (Figure 2a).

In terms of miniCASCO-based cachexia severity, patients with no cachexia had a significantly longer PFS (41 ± 8.5 months) than those with mild (PFS = 19.4 ± 4.2 months; unadjusted HR = 0.3384; 95%CI = 0.1521–0.7526), moderate (17.5 ± 2.7 months; unadjusted HR = 0.2688; 95%CI = 0.1297–0.5569), and severe cachexia (PFS = 12.5 ± 2.7 months; unadjusted HR = 0.4019; 95%CI = 0.1664–0.9706) (*p* = 0.0301) (Figure 2b). 

Analyzing the survival curves based on the evidence of sarcopenia, we demonstrated that patients with no sarcopenia had a longer PFS (28.1 ± 6.9 months) than those with sarcopenia (22.8 ± 4.1 months) (unadjusted HR = 0.4517; 95%CI = 0.2331–0.8754; *p* = 0.0336).

We compared the Kaplan–Meier survival curves based on the mGPS and found that patients with an mGPS of 2 had a significantly lower PFS (13 ± 2.1 months) than those with an mGPS of 1 (16 ± 4.6 months; unadjusted HR = 2.4365; 95%CI = 1.9655–6.1487) and an mGPS of 0 (29.29 ± 3.9 months; unadjusted HR = 2.0682; 95%CI = 1.1389–3.7560) (*p* = 0.0199) (Figure 2c).

The comparison of PFS between patients divided on the basis of the median IL-6 value (6 ng/mL) showed that those with an IL-6 > 6 ng/mL had a significantly lower PFS in comparison to those with an IL-6 ≤6 ng/mL (12.5 ± 2.5 versus 31.3 ± 3.7 months; unadjusted HR = 3.4264; 95%CI = 1.6491–7.1193; *p* = 0.0010) (Figure 2d).

### 3.6. Changes in Inflammatory Status, Nutritional Status, SMI, and miniCASCO Based on the Objective Clinical Response to Treatment

The ANOVA revealed statistically significant differences in the SMI, CRP level, and hemoglobin level among the clinical response categories. Patients with PD had a significantly different decrease in SMI (*p* = 0.003) and hemoglobin level (*p* = 0.011) and a significantly different increase in CRP level (*p* = 0.008) compared to patients with SD, PR, and CR (Appendix A).

## 4. Discussion

Although ICI-based immunotherapy has dramatically improved the prognosis and survival of patients with advanced NSCLC [2,3,36,37,38], there is an unmet need to define potential biomarkers that are suitable for the identification of patients most likely to benefit from immunotherapy [39,40]. Currently, although tumor PD-L1 expression is the only clinically approved and most widely explored predictive biomarker of PD-1/PD-L1 blockade in patients with NSCLC [41], it is an imperfect predictive biomarker [42,43,44]. In the present study, we did not find any correlation between the PD-L1 expression and immunotherapy response, PFS, or OS. This result is consistent with the findings of registrative clinical trials of pembrolizumab, i.e., KEYNOTE-024 [45] and nivolumab [46].

In the context of research, patient characteristics play a crucial role in their response to immunotherapy. Specifically, nutritional status, body composition changes, chronic inflammation, and cachexia have been proposed as essential predictive factors of immune response efficacy [47] and ICI treatment efficacy [32]. Indeed, it is widely reported that a significant proportion of patients with advanced lung cancer have cachexia upon diagnosis [20,21,22]. Consistently, in the present study, most of the patients (53%) with advanced NSCLC had cachexia at the time of diagnosis, and 56.7% of patients had moderate-to-severe cachexia, as defined using the miniCASCO. In this regard, it should be noted that approximately 40% of the patients included in our analysis received ICIs as second-line treatment; therefore, a previous chemotherapy regimen could have influenced the cachectic status. Nevertheless, although in a limited sample size, we did not observe a significant difference in the incidence of cachexia (according to the Fearon criteria) and in the miniCASCO score at enrolment between those patients who received one previous line of treatment and those who did not..

Herein, we identified several patient-related PD-L1-independent parameters that were associated with ICI treatment outcomes. We found a negative association between cachexia (defined based on the criteria proposed by Fearon et al.), miniCASCO-based cachexia severity, and inflammatory status (IL-6 and mGPS) and clinical outcomes (PFS and OS) in patients with advanced NSCLC undergoing ICI monotherapy. Notably, miniCASCO-based cachexia severity and IL-6 levels were independent predictors of clinical response, PFS, and OS. To the best of our knowledge, this is the first study demonstrating that miniCASCO-based cachexia severity is a predictor of clinical outcome in patients with advanced NSCLC undergoing immunotherapy.

Overall, clinical evidence of a relationship between cachexia status and immunotherapy failure in patients with NSCLC is emerging [37]. Several studies have suggested a negative association between cachexia status and ICI treatment response in patients with advanced NSCLC [48,49,50,51,52]. Among the largest studies, Turcott et al. conducted a retrospective study including 300 patients with NSCLC who received any line of immunotherapy and showed that cachexia risk was independently associated with worse PFS and OS, thereby highlighting the role of nutritional assessment in these patients [53]. In 2020, Roch et al., in a study evaluating the association between ICI treatment response and body composition indices, reported an association between cancer cachexia and sarcopenia status with the treatment response rate, PFS, and OS in a group of 122 ICI-treated patients with NSCLC [34]. They found that patients without cachexia were more likely to achieve disease control and had a longer OS than those with cachexia. Moreover, patients with evolving sarcopenia during treatment had a shorter PFS and OS than those without evolving sarcopenia. Miyawaki et al. retrospectively analyzed 157 patients with NSCLC treated with PD-1/PD-L1 inhibitors + chemotherapy or pembrolizumab monotherapy and showed that cancer cachexia, elevated tumor burden, and low PD-L1 expression were independently associated with poor PFS; furthermore, cancer cachexia was significantly associated with poor OS. Moreover, they explored the clinical feasibility of a model integrating tumor burden and cancer cachexia for predicting the therapeutic efficacy of first-line immunotherapy in patients with advanced NSCLC and found that the risk categories based on the immune-related predictive model were significantly associated with PFS and OS [54]. 

Notably, most previous studies on this subject were retrospective, small-scale studies focusing on the relationship between baseline sarcopenia alone (without assessing it in the context of cachexia syndrome and with any parameters of chronic inflammation) and ICI treatment efficacy in patients with advanced NSCLC. The findings revealed that patients with sarcopenia exhibited a poorer outcome than those without sarcopenia [28,32,55,56]. Furthermore, Wang et al. conducted a meta-analysis of nine studies comprising 576 ICI-treated patients with NSCLC and demonstrated that pre-immunotherapy sarcopenia and sarcopenia development or exacerbation during immunotherapy significantly worsened the OS and PFS and reduced the DCR in these patients [57]. Thus, the skeletal muscle mass was recently included in a prognostic score, which independently predicts survival in ICI-treated patients with cancer [58]. Notably, Cortellini et al. [59], in a retrospective analysis including 100 consecutive patients with advanced NSCLC treated with PD-1/PD-L1 checkpoint inhibitors, reported that sarcopenia was significantly associated with a shorter PFS in the univariate analysis (but not in the multivariate analysis) and a shorter OS in both univariate and multivariate analyses. In contrast, they did not find any correlation between SMI and ORR. They concluded that low SMI could not specifically predict the response to immunotherapy. In addition, Nishioka et al., in a retrospective study of 156 patients with NSCLC treated with anti-PD-1/PD-L1 inhibitor monotherapy, showed no relationship between muscle mass and ORR or PFS [31].

Similarly, our study revealed that SMI-defined sarcopenia was not predictive of the PFS, OS, and clinical response to immunotherapy. Sarcopenia seemed not to be an independent predictor of survival, but it is likely to be predictive of survival when in the “vicious circle” with inflammation and malnutrition that characterize cachexia syndrome [60]. In fact, unlike sarcopenia, miniCASCO-based cachexia severity was an independent predictor of PFS, OS, and clinical response. 

Therefore, our prospective study findings corroborate previous retrospective study findings regarding the adverse effects of cachexia on ICI treatment outcomes. Although most previous studies hypothesized that sarcopenia might reflect an increased catabolic activity associated with most aggressive tumors (which involves inflammation and related muscle wasting), they did not assess the levels of inflammatory variables (such as CRP and proinflammatory cytokines) as potential catabolic drivers. In contrast to these studies, we used a multifaceted tool for cancer cachexia status definition and severity classification, the miniCASCO, which incorporates pretreatment weight loss, BMI, body composition, SMI, inflammatory/immunological parameters, and functional and quality of life indices. 

Additionally, despite the small sample size, our study demonstrated a relationship between inflammatory indexes (especially CRP level, IL-6 level, and mGPS) and clinical response to ICI, PFS, and OS. Notably, the IL-6 level was an independent predictive factor of clinical response, PFS, and OS. Indeed, IL-6 is the key cytokine involved in the pathogenesis of cachexia-related changes in energy metabolism, nutritional status, and body composition, as well as related signs and symptoms such as anorexia, anemia, and fatigue [23,61]. Notably, IL-6 is a major mediator of muscle mass wasting by interfering with several muscle catabolic and anabolic pathways [62,63]. IL-6-driven inflammation is strongly correlated with an increase in the levels of CRP and other acute-phase proteins, such as fibrinogen, by direct transcription induction via the liver in a dose- and time-dependent manner [64]. Acute-phase proteins promote immunodepression and have been associated with, together with IL-6, leukocytosis and lymphopenia [65]. Thereby, an increase in the levels of inflammatory markers, such as CRP [66] and NLR [67], has been significantly correlated with cancer cachexia and sarcopenia [56,68]. The prognostic role of the CRP level has also been assessed in combination with the albumin level in the mGPS, an inflammatory/nutritional index, which is the most validated prognostic index in patients with cancer [69]. The mGPS has an inverse association with sarcopenia and cachexia. Moreover, the mGPS is one of the most important host-related prognostic parameters in patients with lung cancer [70,71,72].

In addition to the mGPS, other scores based on local and systemic host-tumor interactions that are closely related to the immunological and nutritional status of patients, such as the neutrophil-to-lymphocyte ratio (NLR), platelet-to-lymphocyte ratio, C-reactive protein (CRP)–albumin ratio (CAR), prognostic nutritional index (PNI), and advanced lung cancer inflammation index (ALI), systemic immune-inflammation index (SII), and lung immunoprognostic score (derived from the neutrophil count [neutrophil-lymphocyte ratio], have been shown to be associated with prognosis in ICI-treated patients with NSCLC [4,5,6,7,8,9,10,73,74,75,76]. In our study, NLR did not demonstrate any predictive or prognostic role. In contrast, in the last decade, several studies have supported the prognostic role of NLR in immunotherapy-treated patients with NSCLC [75,76,77,78,79]; in addition, a meta-analysis by Jiang et al. supported the negative prognostic role of high NLR in immunotherapy-treated patients with NSCLC [10]. Notably, NLR seems to predict outcomes independent of treatment modality [80]. Although we did not demonstrate a prognostic role of NLR, we observed a significant inverse association between absolute neutrophil count and disease control upon univariate regression analysis. Different mechanisms may link peripheral leukocytosis with prognosis and tumor progression [81]; furthermore, the blood neutrophil count, identified by NLR, reportedly has a direct link with the intratumoral neutrophil count, and thus may potentially compromise the antitumor immune response [82]. Moreover, we found that the absolute lymphocyte count had a significant positive correlation with both PFS and OS; however, it was not an independent predictor in the multivariate analysis. A low circulating lymphocyte count is known to reflect an impairment of cell-mediated immunity and is associated with the prognosis in patients with solid tumors [83]. In this regard, previous studies published by our group showed that tumor-associated lymphocytes from pleural neoplastic effusions of patients with lung cancer are defective in terms of proliferative response and immune functions, although they can release high amounts of various cytokines, especially IL-6 [84,85]. Moreover, the defective capacity of lymphocytes in patients with cancer has been associated with increased levels of proinflammatory cytokines and acute-phase proteins, both in tumor effusions and in peripheral blood [65,85]. The absolute lymphocyte count is not only an indirect measure of the antineoplastic immune response but also a known index of impaired nutritional status and altered energy metabolism, which, in turn, can affect lymphocyte activation, proliferative capacity, and functional status [86]. The literature is replete with evidence that nutritional status is crucial for immune cell function, and undernutrition is associated with immunosuppression [87].

Our results showing that both IL-6 and cachexia severity were independent predictive factors of PFS and OS in patients with NSCLC under ICI support the hypothesis that the negative associations between cachexia and ICI efficacy can mainly be attributed to chronic inflammation, as we reported in a recent review [24]. Chronic inflammation, with related changes in nutritional status, body composition, and energy metabolism, negatively affects the functioning or efficiency of the immune system via several pathways [27], thus influencing the response to immunotherapy [88,89,90]. The same cytokines that function as key mediators of cachexia, especially IL-6, can compromise and directly suppress several antitumor immune functions [26,27,91,92,93,94,95,96,97], while their inhibition can reinvigorate the antitumor response and exert synergistic effects with the blockade of the PD-1/PD-L1 axis [92]. Furthermore, chronic inflammation may cause immunosuppression by inducing the proliferation of immunosuppressive cells, such as myeloid-derived suppressor cells [98,99], which, in turn, can overcome the inhibition of immunosuppression mediated by ICIs within the tumor microenvironment, thus influencing their therapeutic effect [100,101]. Chronic inflammation is known to be also associated with increased production of reactive oxygen species and oxidative stress [23], which may suppress T cell function and induce specific alterations in T-cell-receptor signaling, thereby reducing the antigen-mediated response of effector T cells [27].

Additionally, the complex metabolic abnormalities associated with cancer cachexia, mainly mediated by inflammation, reportedly contribute to antitumor activity suppression and a reduction in the response to immunotherapy [93,102]. During T cell activation, T cells have specific energy requirements (such as enhanced glycolysis and efficient iron metabolism) to proliferate and generate an effective immune response [27,103,104,105]. In cancer cachexia, systemic metabolic derangements, anemia, and anorexia trigger an impairment of nutritional intake and utilization of energy substrates that are fundamental for the main lymphocyte energy metabolic pathways [96,106], leading to T cell defective activation (anergy) and exhaustion, which is characterized by a progressive loss of T cell function [27]. Moreover, T cell energetic and anabolic processes can be hampered by the IL-6-mediated inhibition of the PI3K/Akt/mTOR pathway [107]. Therefore, the same molecular and biological inflammatory-mediated pathways that regulate the pathogenesis of cachexia can negatively affect a wide spectrum of immune responses; thus, the vicious circle of chronic inflammation and cachexia/sarcopenia in patients with NSCLC can negatively influence the antitumor immune response to ICI immunotherapy.

Of note in the present study, we demonstrated that patients who encountered PD during ICI treatment had a decrease in SMI and an increase in CRP during treatment. These findings are consistent with those of Roch et al. [34], who reported that evolving sarcopenia, defined as a reduction in SMI of >5% during immunotherapy, is associated with adverse survival outcomes. Similarly, in 2022, Shijubou et al. [108], in a retrospective study on 38 pembrolizumab-treated patients with advanced NSCLC, found that weight loss of >5% after treatment initiation is an important negative prognostic factor; hence, they concluded that weight maintenance might be important for good ICI treatment efficacy. Likewise, Degens et al., in a population of 106 patients with advanced NSCLC receiving second-line immunotherapy with nivolumab, found that weight loss >2% at week six of treatment is an independent predictor for poor OS; nevertheless, they did not demonstrate the effect of skeletal muscle loss on OS [109].

Additionally, we found that patients with PD during ICI treatment had a significantly different decrease in hemoglobin in comparison to patients with CR/PR and SD. Cancer anemia is a known consequence of chronic inflammation with associated functional iron deficiency, and it is associated with changes in nutritional status as well previously described by us [110]. Anemia with low oxygen and low iron availability, in turn, blunts oxidative phosphorylation and tricarboxylic acid cycle activities, thus inducing energy defects in T cells and consequent impairment of the T-cell-mediated immune antineoplastic response [27]. Therefore, it can be hypothesized that in addition to the pretreatment evaluation of cachexia syndrome, dynamic bodyweight loss or evolving sarcopenia early during ICIs treatment may be associated with poor outcomes. In fact, these parameters reflect an ongoing catabolic process, which leads to an inhibition of many features of the antitumor immune response, thereby suppressing immunotherapy efficacy [34]. This corroborates the findings of our recent review that cachexia may be reversible if the clinical response to anticancer therapy decreases the main catabolic drivers, which would comprise the tumor and the tumor-related inflammation [23].

Additionally, our finding that cachexia can influence ICI efficacy could serve as a reference for further clinical studies exploring the efficacy of combining immunotherapy with anti-cachexia drugs to improve ICI efficacy. In particular, an early multimodal approach based on nutritional and anti-inflammatory strategies could improve (in addition to the key components of cachexia) the key aspects of the antitumor immune response, thereby improving the ICI immunotherapy effectiveness in a clinical setting. In this regard, since 2000, we have conducted studies evaluating the efficacy of a combination treatment approach consisting of weekly chemotherapy with cisplatin and epirubicin with immunotherapy (recombinant IL-2) and anti-inflammatory (medroxyprogesterone acetate) and antioxidant agents in patients with advanced (stage IIIB-IV) lung cancer with cancer-related anorexia and cachexia [111].

To the best of our knowledge, this is one of the few prospective studies investigating the effect of cancer cachexia on ICI treatment outcomes and the first using a multidimensional tool for defining cachexia and its severity. Another strength of our study is that our sample population consisted exclusively of patients with NSCLC who received PD-1/PD-L1 monotherapy.

However, the present study has some limitations. The sample size was relatively small, although the study population was homogeneous in terms of the tumor stage. Furthermore, different ICIs were used in this study, and we included patients receiving different lines of ICI treatment. Moreover, the miniCASCO is not routinely used in clinical oncology; therefore, large-scale multicenter studies are warranted for the external validation of these results.

## 5. Conclusions

This prospective, longitudinal, observational study demonstrated how an integrated analysis of cancer cachexia and its severity, including the parameters of body composition, nutritional status, and inflammatory status, merge as a key parameter in relation to clinical response and outcomes in ICI-treated NSCLC patients. Our results support the evidence that cachexia could be considered an independent unfavorable prognostic factor and a predictor of a worse ICI treatment response; however, large-scale studies are needed to confirm the present findings. Future studies should consider cachexia and its severity as additional classification factors in the design of immunotherapy-based trials. In fact, careful assessment for cachexia and related abnormalities prior to ICI treatment initiation could help to identify patients who are more likely to achieve a better treatment response to PD-1/PD-L1 inhibitors. This could be integrated into clinical practice since it is based on feasible clinical, imaging, and laboratory analyses yet included in the routine evaluation of NSCLC cancer patients. Such assessment could fit with and integrate into other existing models and available inflammatory/nutritional scores such as NLR, PNI, and CAR, which are calculated using the same parameters that are useful to diagnose and classify cancer cachexia. These parameters could be weighted and integrated to calculate a predictive score/grading to be tested in future prospective clinical studies.

Moreover, the use of the miniCASCO questionnaire, which allows for early recognition of cachexia, could be beneficial for patients with NSCLC who are candidates for immunotherapy. Using integrated early targeted supportive care in patient treatment can counteract cancer cachexia and its related abnormalities; this might be the best strategy to enhance ICI treatment efficacy.

Further clinical trials are required to expand the knowledge in this field by investigating the different components of the anti-cancer immune response and their dynamicity during treatment. In addition, the role of nutritional/inflammatory indices as well as cachexia staging tools, such as miniCASCO, should be further investigated in other prospective studies to confirm their predictive and prognostic roles and the ability to identify patients who could benefit from ancillary therapeutic strategies to ameliorate their clinical outcomes and quality of life.

## Figures and Tables

**Figure 1 cancers-15-01076-f001:**
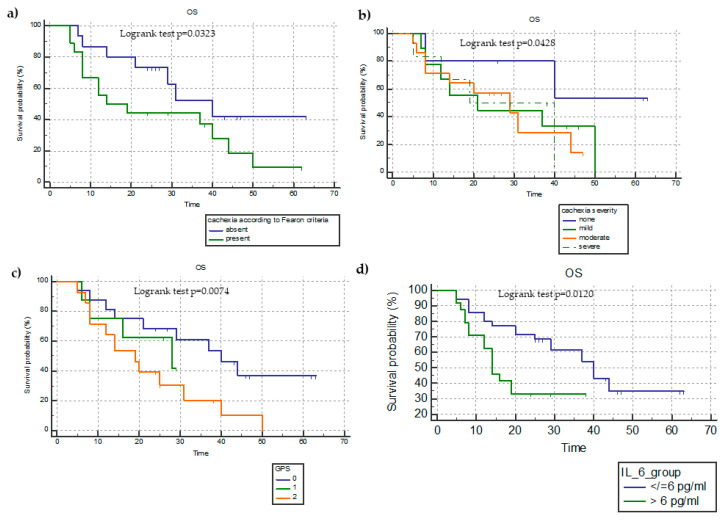
Kaplan–Meyer survival analysis of overall survival (OS) in terms of the presence of cachexia (**a**), miniCASCO-based cachexia severity (**b**), mGPS categories (**c**), and median IL-6 level (**d**).

**Figure 2 cancers-15-01076-f002:**
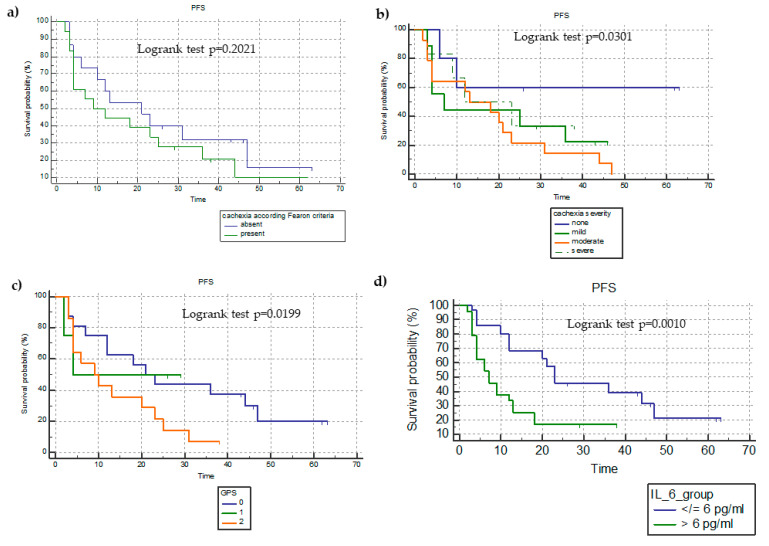
Kaplan–Meyer survival analysis of progression-free survival (PFS) in terms of the presence of cachexia (**a**), miniCASCO-based cachexia severity (**b**), GPS categories (**c**), and IL-6 median level (**d**).

**Table 1 cancers-15-01076-t001:** Patients’ anthropometric and clinical characteristics at baseline.

Parameters	Mean ± SD (Range)	No. (%)
Enrolled patients		74 (100)
Males/females		54/20 (73/27)
Age (years)	69.3 ± 11.3 (47–88)	
Weight (kg):	68.9 ± 12.4 (46–95)	
Height (cm):	164.1 ± 8.7 (151–185)	
Stage		
IV		74 (100)
Histology		
Adenocarcinoma		57 (77)
Squamocellular		17 (23)
PD-L1 expression		
<1%		10 (13.5)
1–50%		22 (29.8)
>50%		32 (43.2)
N.V.		10 (13.5)
ECOG–PS		
0		10 (13.5)
1		39 (52.7)
2		25 (33.8)
Treatment		
Nivolumab		16 (43.2)
Pembrolizumab		21 (56.8)
Previous line		32 (43)

Data are reported as the mean ± standard deviation (SD) for continuous variables or number and percentage for categorical variables. Abbreviations: PD-L, programmed death-ligand; ECOG PS, Eastern Cooperative Oncology Group Performance Status.

**Table 3 cancers-15-01076-t003:** Logistic regression analysis between the parameters of cachexia, body composition/nutritional status, inflammatory status, and miniCASCO and the clinical response (disease control versus progressive disease).

Parameters	Regression Coefficient	OR	95%CI	*p*-Value
Baseline weigh loss %	−0.0346	0.966	0.8588–1.0866	0.5646
Cachexia according to the standard criteria [24]	−0.7885	0.4545	0.1602–1.2893	0.1383
LBM	−0.0158	0.9843	0.9326–1.0389	0.5658
SMI	0.0129	0.9871	0.9493–1.0265	0.5154
C-reactive protein	−0.672	0.9971	0.8164–0.9848	**0.0227**
IL-6	−0.0799	0.9232	0.8762–0.9728	**0.0027**
Neutrophil count	−0.00022	0.9998	0.9996–1.0000	**0.0403**
Lymphocyte count	−0.008	0.9992	0.9983–1.0001	0.0757
NLR	−0.0819	0.9213	0.7460–1.1378	0.4467
mGPS	−0.4	0.6701	0.3906–1.1498	0.1462
Albumin	−0.2878	0.7499	0.2340–2.4034	0.6282
Hemoglobin	0.1545	1.1671	0.8953–1.5214	0.2533
MiniCASCO score	−0.1651	0.9035	0.6766–0.9845	**0.0186**
Anorexia subscale	0.1607	1.1743	1.0056–1.3713	**0.0423**
IMD subscale	−0.0466	0.9545	0.8565–0.9988	**0.0469**
BWC subscale	−0.0187	0.9814	0.9318–1.0337	0.4789
PHP subscale	−0.0164	0.9659	0.8847–1.002	0.2228
QoL subscale	−0.1015	0.9035	0.6766–1.2065	0.4917
Cachexia severity	−0.1511	0.8598	0.4957–1.4914	0.5908

Significant *p*-values (<0.05) are reported in bold. Abbreviations: LBM, lean body mass; SMI, skeletal mass index; IL, interleukin; NLR, neutrophil-to-lymphocyte ratio; mGPS, modified Glasgow Prognostic Score; BWC, bodyweight composition/change; IMD, inflammation/metabolic disturbances/immunosuppression; PHP, physical performance (PHP); QoL, quality of life.

## Data Availability

Original clinical, laboratory, and instrumental data can be found in the patient chart archived at the Department of Medical Oncology and are available on request from the corresponding author.

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
