# Peer review of "Effect of Cancer-Related Cachexia and Associated Changes in Nutritional Status, Inflammatory Status, and Muscle Mass on Immunotherapy Efficacy and Survival in Patients with Advanced Non-Small Cell Lung Cancer"

_cancers, 2023, doi:10.3390/cancers15041076_

Round 1
Reviewer 1 Report
The article by Madeddu et al. designed a perspective observational study to evaluate the prognostic and predictive role of cachexia, with its related changes of inflammatory, immunological, and nutritional parameters, on survival and clinical response to ICI in patients with advanced NSCLC. Their results suggest that cachexia can be an independent unfavorable prognostic and predictive factor. Although the results were very important, there seems some limitation and many flaws in the interpretation and statistical methos.
1. Abstract and throughout the manuscript:
The meaning of “clinical response” is unclear. In “2.3 Outcome”, the meaning of “objective response” and “objective clinical response” is unclear.
2. 2.3 Outcome: When was the “first computed tomography” performed? When was the “first instrumental evaluation of objective clinical response” (2.4 Collection of clinical and laboratory data)? These data seem mandatory because “clinical response” was one of the important endpoint of the study.
3. 3.2 Association between classic ...
There are “r” coefficients, which seems to be results from the Pearson’s correlation analysis. However, there is no statement regarding the Pearson’s correlation in the Methods-Statistical analysis section. Furthermore, “r” means that it is a comparison between two continuous variables. However, it is unclear how the authors presented “clinical response” as a continuous variable. It is also the case in “3.3 Association of cachexia ...” (the meaning of “clinical response” on the 1st line and “objective clinical response” on the last line is unclear).
4. 3.3 Association of cachexia ...
There is a statement of multivariate regression analysis, but the detail of the analysis is unclear. In the Methods, multivariate regression was explained as logistic regression. Is it the case? If so, it is quite unnatural because univariate analysis was not a logistic regression but Pearson’s analysis with r.
5. 3.4 Association of cachexia status, miniCASCO, ...
In the assessment of survival, the authors used Cox regression in “3.5 Cox regression survival analysis ...”. What is the meaning of the section “3.4”? The comparison between some continuous variables and PFS (also with Pearson’s r) does not make sense. In the Cox regression in 3.5, there are univariate analyses only. In the multivariate analysis for survival, multivariate Cox regression is needed, partly because it can consider the effect of censoring.
6. 3.5 Cox regression survival ...
“The OS analysis based on GPS showed that patients with GPS 0 had a higher mean OS”
How the author consider it was “higher”? The Kaplan-Meier method can only find the significant difference among the subgroups.
7. Figure 3 and 4
Indicate the HR as adjusted or unadjusted (crude) HR clearly.
Minor points:
1. Simple Summary: “(ICI)-based” instead of “(ICI)I-based”
2. Simple Summary: “a prospective observational study” instead of “a perspective observational study”
3. Abstract: “a prospective study” instead of “a perspective study”
4. 2.1 Study design: “ORR” should pe spelled-out.
5. 4. Discussion: line 424-426
Avoid duplication of statement between the Introduction and the Discussion.
6. Discussion section is too long. Please concentrate on the discussion regarding the results of the present study.
Author Response
Point-by-point reply to Reviewer comments
Reviewer 1
Comments and Suggestions for Authors
The article by Madeddu et al. designed a perspective observational study to evaluate the prognostic and predictive role of cachexia, with its related changes of inflammatory, immunological, and nutritional parameters, on survival and clinical response to ICI in patients with advanced NSCLC. Their results suggest that cachexia can be an independent unfavorable prognostic and predictive factor. Although the results were very important, there seems some limitation and many flaws in the interpretation and statistical methos.
Reply: Dear reviewer, thank you for your positive feedback on our paper and for your comments. I have substantially revised the manuscript according to your indications.
- Abstract and throughout the manuscript:
The meaning of “clinical response” is unclear. In “2.3 Outcome”, the meaning of “objective response” and “objective clinical response” is unclear.
Reply: I have substantially revised the paragraph by better specified the meaning of objective clinical response in the abstract and in the paragraph "2.3 Outcome”. Indeed, I have specified that I assessed the association of different parameters with disease control rate. I have revised the paragraph 2.3 to better clarify the exact outcomes and parameters assessed in the different analysis and presented in the results (page… lines..). Moreover, I have also improved the description of the methods and criteria used for the clinical response evaluation in the paragraph 2.1 (page…, lines…).
- 2.3 Outcome: When was the “first computed tomography” performed? When was the “first instrumental evaluation of objective clinical response” (2.4 Collection of clinical and laboratory data)? These data seem mandatory because “clinical response” was one of the important endpoint of the study.
Reply: I have specified when the first instrumental evaluation of objective clinical response was performed, i.e., at week eight according to international validated RECIST criteria for the evaluation of objective clinical response during immunotherapy. I have specified this point in the paragraph 2.4 as well as in the paragraph 2.1 and 2.3. I have also specified for better clarity the timing for performing baseline CT evaluation (i.e., no more than one month earlier the start of immunotherapy): see paragraph 2.2 and paragraph 2.4.
- 3.2 Association between classic ...
There are “r” coefficients, which seems to be results from the Pearson’s correlation analysis. However, there is no statement regarding the Pearson’s correlation in the Methods-Statistical analysis section. Furthermore, “r” means that it is a comparison between two continuous variables. However, it is unclear how the authors presented “clinical response” as a continuous variable. It is also the case in “3.3 Association of cachexia ...” (the meaning of “clinical response” on the 1st line and “objective clinical response” on the last line is unclear).
Reply: I have added in the Methods Statistical analysis section the method used for correlation analysis. I have clarified that I have used the Pearson or Spearman correlation analysis, that results in the “r” factor for correlation between continuous variables (for parametric or not parametric variables, respectively). Moreover, I agree with your comments regarding clinical response, which is not a continuous but a categorical variable, as correctly you noted. Therefore, I have corrected the text in the methods-statistical analysis section, where I have specified that we used logistic regression analysis (“Probabilities and odds ratios (OR) of a change in categorically dependent variables, namely disease control versus progressive disease at the eighth week, conditional on the values of independent covariables were analyzed in a logistic regression model. Variables associated with a p value < 0.5 in univariate analyses were included into multivariate models, multivariate regression analysis (stepwise method), considering the clinical response as dependent variables.”). Consistently, I have revised the title and the content of the paragraph 3.2 to better illustrate the results according to the values obtained by the logistic regression analysis. I have also clarified what we mean for clinical response by writing more precisely the following : “objective clinical response categories (i.e., objective clinical response categories, i.e., disease control versus progressive disease,…..” (page). The same corrections have been inserted in the paragraph 3.3.
- 3.3 Association of cachexia ...
There is a statement of multivariate regression analysis, but the detail of the analysis is unclear. In the Methods, multivariate regression was explained as logistic regression. Is it the case? If so, it is quite unnatural because univariate analysis was not a logistic regression but Pearson’s analysis with r.
Reply: in accordance with your right correction above, as indicated in the statistical methods I used logistic regression analysis for categorical variables, i.e., objective clinical response. See the revised text in the Methods Section: “Correlation between continuous variables was assessed by Pearson or Spearman correlation analysis for parametric or not parametric data, respectively. Probabilities and odds ratios (OR) of a change in categorically dependent variables, namely disease control versus progressive disease at the eighth week, conditional on the values of independent covariables were analyzed in a logistic regression model. Variables associated with a p value < 0.5 in univariate analyses were included into multivariate models, multivariate regression analysis (stepwise method), considering the clinical response as dependent variables.”. Therefore, I have corrected the table and the results accordingly, by specifying the regression coefficient value, the OR value with its 95%CI interval and the p value. See revised paragraph 3.3
- 3.4 Association of cachexia status, miniCASCO, ...
In the assessment of survival, the authors used Cox regression in “3.5 Cox regression survival analysis ...”. What is the meaning of the section “3.4”? The comparison between some continuous variables and PFS (also with Pearson’s r) does not make sense. In the Cox regression in 3.5, there are univariate analyses only. In the multivariate analysis for survival, multivariate Cox regression is needed, partly because it can consider the effect of censoring.
Reply: In the first version of the manuscript, Section 3.4 report the results of the association analysis (performed by Spearman or Pearson for continuous variables, and by ANOVA or Kruskal Wallis for categorical variables) between PFS/OS and the different variables assessed. Moreover, according with your comment, where you judge this paragraph of less importance and not useful, in comparison to COX regression analysis, I have moved it as an Appendix file. In the appendix, for more clarity, I have revised the description of the results and distinguished between continuous and categorical variables.
As regard COX regression analysis, I have added univariate and multivariate COX regression analysis in order to assess the determinants of PFS and OS in a different specific paragraph (Paragraph 3.4 of the revised version. I have also better specified the methods used for survival analysis in the methods section as follows: “Survival analyses were calculated using the Kaplan-Meier (log rank) test. We used the Cox proportional hazards model for univariate and multivariate analysis to calculate hazard ratios (HRs). Possible factors identified by univariate analysis were evaluated separately for PFS and OS using multivariate Cox regression analysis and the stepwise method to determine independent predictors of PFS and OS rates.” The revised version of the COX regression analysis is reported in the Paragraph 3.4 of the revised version.
- 3.5 Cox regression survival ...
“The OS analysis based on GPS showed that patients with GPS 0 had a higher mean OS”
How the author consider it was “higher”? The Kaplan-Meier method can only find the significant difference among the subgroups.
Reply: I have better explained that the comparison between subgroups was performed by Kaplan-Meier method (therefore, “higher” and the reported significant differences were established by Kaplan-Meier method. For better clarity I have titled this paragraph “Kaplan-Meier survival analysis of PFS and OS between ……..”. I have separated this data from the results of univariate e multivariate COX regression analysis that I have now reported in a specific paragraph (paragraph 3.4 of the revised version).
- Figure 3 and 4
Indicate the HR as adjusted or unadjusted (crude) HR clearly.
Reply: The statistical analysis and the results reported the unadjusted HR. I have reported the complete data including HR, 95%CI and p value, in the text of the results for better clarity (Paragraph 3.5 of the revised version). I have revised the Figures including significant results.
Minor points:
- Simple Summary: “(ICI)-based” instead of “(ICI)I-based”
Reply: thank you. I have corrected the text as indicated by you both in the Simple Summary (line 8) and in the Abstract (line 31).
- Simple Summary: “a prospective observational study” instead of “a perspective observational study”
Reply: I have corrected “perspective” into “prospective” in the Simple Summary (line 25)
- Abstract: “a prospective study” instead of “a perspective study”
Reply: I have corrected “perspective” into “prospective” in the Abstract (line 36).
- 1 Study design: “ORR” should pe spelled-out.
Reply: I have spelled out ORR as follows: “……..Overall Response Rate (ORR)”
- 4. Discussion: line 424-426
Avoid duplication of statement between the Introduction and the Discussion.
Reply: I have checked carefully and removed the duplication of statement between the Introduction and the Discussion.
- Discussion section is too long. Please concentrate on the discussion regarding the results of the present study.
Reply: I have shortened the discussion section of more than 500 words and concentrate it on the discussion regarding the results of the study. Overall, I have shortened Introduction and Discussion of about 1000 words.

Reviewer 2 Report
This prospective study details on the role of sarcopenia and the relationship with ICI’s in patients with NSCLC. Although the paper is interesting and relevant, I have several specific comments as outlined per section below:
Abstract
1. No further comments.
Introduction
1. The introduction itself is too lengthy and can be shortened. Specifically, the authors could detail on the CACS syndrome earlier in their introduction and move some earlier segments to the discussion.
2. There is an elaborate discussion of previous studies in the introduction which should either be shortened or moved to the discussion section altogether
3. The introduction states a “perspective” cohort study which should be a “prospective” cohort study
Methods
1. What are the standard doses of immunotherapy used in the study? Did these vary by patient? What was the treatment interval?
2. The study was open for approximately 6 years but only 74 patients were enrolled. What were the selection criteria/exclusion criteria, how many patients were screened and how many were eventually approached?
3. The calculations for assessment of skeletal mass index as presented in the methods could be added as an appendix instead of the calculations as currently presented in the paper
4. Were patients treated with both nivolumab and pembroluzimab or either one of these drugs? This should be clarified in the methods
Results
1. Approximately 40% of patients were treated with a previous line of treatment. What was this form of treatment and how may this have impacted the results? I can imagine that previous treatment with chemotherapy, surgery and/or radiation therapy may also significantly impact cachexia scores.
2. How long was the follow-up period?
3. What happened after disease progression on treatment with immunotherapy? Were patients excluded from the current study or were they treated with a different modality and were these results included?
4. There is a total of 10 different tables + figures in the paper. I think the authors would to well to critically assess their results and try to reduce the total number of tables + figures to 5 and streamline their results section. The results section is currently difficult to interpret and could be shortened by only presenting the most relevant outcomes and selecting some results to be moved to an appendix.
Discussion
1. The current study only included patients with a performance status of ECOG 0, 1 or 2. I feel that cachexia, or cachexia syndromes, are closely linked to the ECOG performance score as demonstrated in previous studies. How do the authors relate their current findings to the performance score in light of only having included patients with ECOG 0, 1 or 2?
2. The discussion section is quite lengthy and details on a fair amount of previous research in overlap with the introduction. I strongly feel that both sections could be shortened to present only the most relevant previous literature to the current study.
3. The authors suggest the development of a predictive model to better understand response to ICI’s. How would such a model be incorporated in clinical practice and how would it “fit” within existing models?
Author Response
Point-by-point reply to Reviewer comments
Reviewer 2
Comments and Suggestions for Authors
This prospective study details on the role of sarcopenia and the relationship with ICI’s in patients with NSCLC. Although the paper is interesting and relevant, I have several specific comments as outlined per section below:
Reply: thank you for your positive evaluation and for your major comments. I have revised the manuscript accordingly and I hope that the revised version satisfies your requests and addresses your concerns and that the manuscript can result improved.
Abstract
- No further comments.
Introduction
- The introduction itself is too lengthy and can be shortened. Specifically, the authors could detail on the CACS syndrome earlier in their introduction and move some earlier segments to the discussion.
Reply: I have shortened the introduction of about 400 words. In particular, I have moved earlier the part on CACS and moved some earlier parts to the Discussion.
- There is an elaborate discussion of previous studies in the introduction which should either be shortened or moved to the discussion section altogether
Reply: I have shortened the discussion of previous studies in the Introduction and moved them to the Discussion section.
- The introduction states a “perspective” cohort study which should be a “prospective” cohort study
Reply. I have corrected “perspective” into “prospective”.
Methods
- What are the standard doses of immunotherapy used in the study? Did these vary by patient? What was the treatment interval?
Reply: The patients have been treated all with the same standard dose and the conventional regimen approved for nivolumab and pembrolizumab. I have specified these data in the text (page 3, lines 118-121).
- The study was open for approximately 6 years but only 74 patients were enrolled. What were the selection criteria/exclusion criteria, how many patients were screened and how many were eventually approached?
Reply: First of all I have checked again the time period I cover for patients evaluation for the study and the period was from March 2 2017 to August 12 2021 (there was an error in typing the text). As regard the inclusion/exclusion criteria, I have already specified the inclusion/exclusion criteria in the text (page 4, lines 149-169). Moreover, as I agree with you and your comment, in order to clarify how many patients were screened, how many excluded and the reasons for exclusion, and how many approached and included in the analysis, in the revised version I have added a flow diagram of the study (see page 6, lines 260-266, and Figure 1 of the revised version).
- The calculations for assessment of skeletal mass index as presented in the methods could be added as an appendix instead of the calculations as currently presented in the paper
Reply: I have added the calculations for the assessment of skeletal muscle as an appendix and removed them from the Method Section of the main text.
- Were patients treated with both nivolumab and pembrolizumab or either one of these drugs? This should be clarified in the methods
Reply: Please note that yet in the original version of the manuscript at lines 141-142 it was specified that we enrolled patients who were “eligible to receive monotherapy with nivolumab or pembrolizumab”. Therefore, patients received nivolumab or pembrolizumab monotherapy (page 6, line 118).
Results
- Approximately 40% of patients were treated with a previous line of treatment. What was this form of treatment and how may this have impacted the results? I can imagine that previous treatment with chemotherapy, surgery and/or radiation therapy may also significantly impact cachexia scores.
Reply: I have specified in the text the previous chemotherapy regimens; I have also specified that all patients received one previous line of treatment (lines 282-286, page 7). Surely previously line of treatments can influence prognosis and cachexia status. However, cachexia status and all parameters have been assessed before enrolment in the present study, thus reflecting the actual status of the patients at the time of enrolment. The evaluation of the impact of previous treatments is out of the scope of the study. Moreover, following your comment I have calculated how many patients were cachectic between not-previously treated patients and previously treated patients: they were 54% and 60%, respectively. The difference was not significantly different. Nevertheless, taking into account your comment I have added a comment (one sentence) in the Discussion section where I have reported the incidence of cachexia in our patient population (see lines 479-484, page 11): “At this regard, it should be noted that about 40% of patients included in our analysis received ICIs as second-line treatment; therefore, a previous chemotherapy regimen could have influenced the cachectic status. Nevertheless, although in a limited sample size, we did not observe a significant difference in the incidence of cachexia (according to Fearon criteria) at enrolment between patients who received one previous line of treatment and those who did not (data not shown)” .
- How long was the follow-up period?
Reply: The follow up period was until death since these are all metastatic patients at diagnosis. I have added the range of follow-up period in the Results (paragraph 3.1).
- What happened after disease progression on treatment with immunotherapy? Were patients excluded from the current study or were they treated with a different modality and were these results included?
Reply: Thank you for your question. After disease progression with immunotherapy the patients were treated based on general condition with a new treatment if eligible or with best supportive care. The results were included in the analysis of OS (obviously not in the analysis of PFS).
- There is a total of 10 different tables + figures in the paper. I think the authors would to well to critically assess their results and try to reduce the total number of tables + figures to 5 and streamline their results section. The results section is currently difficult to interpret and could be shortened by only presenting the most relevant outcomes and selecting some results to be moved to an appendix.
Reply: I have reduced the total number of tables to 4 and figures the 2 (including the flow diagram of the study) and moved the remaining to an appendix. Also, the results have been shortened and some less relevant data (according to your comments) have been moved as an appendix, in particular the entire paragraph 3.4 of the previous version.
Discussion
- The current study only included patients with a performance status of ECOG 0, 1 or 2. I feel that cachexia, or cachexia syndromes, are closely linked to the ECOG performance score as demonstrated in previous studies. How do the authors relate their current findings to the performance score in light of only having included patients with ECOG 0, 1 or 2?
Reply: Thank you for your comment. Firstly, the inclusion of patients with performance status ECOG 0,1,or 2 was determined by the present criteria established by regulatory pharmaceutical guidelines, as well as international guidelines (e.g. NCCN guidelines), for the eligibility to treatment with anti-PD1 inhibitors nivolumab and pembrolizumab. Therefore, since our study was carried out in a population of NSCLC patients eligible for anti-PD1 treatment, where we assessed the impact of cachexia and related parameters on clinical outcome, we have to include only patients with performance status ranging from 0 to 2. Moreover, it should be noted that cachexia diagnosis is now based on the Fearon criteria that use the percentage of weight loss eventually corrected for BMI or presence of sarcopenia [i.e., 5% weight loss or ≥2% weight loss in individuals already showing decreases in body weight and height (body mass index [BMI] < 20 kg/m2) or skeletal muscle mass (sarcopenia)]. Therefore, the diagnosis of cachexia did not automatically and always imply a performance status >2, but can be observed in patients that often have a performance status of 1 and 2, not necessarily 3 or 4. Moreover, it should take into account that cachexia evolves through different stages from reversible to irreversible or refractory cachexia (as defined by international consensus meeting and published in Lancet Oncol. 2011 May;12(5):489-95. doi: 10.1016/S1470-2045(10)70218-7. In this publication is defined how only refractory cachexia is characterized by low performance status: “refractory cachexia is characterized by a low performance status (WHO score 3 or 4) and a life expectancy of less than 3 months”. Therefore, it is widely recognized that cachexia, or cachexia syndromes, not necessarily is closely linked to a poor ECOG performance score (i.e., ECOG 3 and 4). As regard your comment on the relation of our findings with performance status, I did not found any predictive role of performance status in regard to clinical response, PFS and OS. Obviously, the inclusion of only Performance status 0 to 2 can have influenced the absence of impact of performance status on PFS and OS, although this is not a primary or secondary outcome of our study. Nevertheless, I have commented the lack of association in the results by indicating that “The lack of association between PS and clinical outcomes could have been influenced by the inclusion of only patients with ECOG PS 0-2 as requested by the established regulatory criteria for the eligibility to anti-PD1 therapy with nivolumab and pembrolizumab” (page 12, lines 347-349).
- The discussion section is quite lengthy and details on a fair amount of previous research in overlap with the introduction. I strongly feel that both sections could be shortened to present only the most relevant previous literature to the current study.
Reply: in accordance with your punctual comment, I have shortened both the introduction and the Discussion section and eliminated the overlapping parts (overall I have reduced the text of more than 1000 words).
- The authors suggest the development of a predictive model to better understand response to ICI’s. How would such a model be incorporated in clinical practice and how would it “fit” within existing models?
Reply: I have added the following sentences in the Discussion to explain as our assessment can be incorporated in clinical practice and fit with other existing models: “This could be integrated in the clinical practice since it is based on easily and feasible clinical, imaging and laboratory analysis, yet included in the routinary evaluation of NSCLC cancer patients. Such assessment could fit and integrate other existing models and available inflammatory/nutritional scores such as NLR, PNI, CAR, which use for calculation the same parameters useful to diagnose and classify cancer cachexia. These parameters could be weighted and integrated to calculate a predictive score/grading to better understand response to ICI’s to be tested in future perspective clinical studies.” (page 17, lines 740-746).
English has been revised by an English Editing Service (I have attached the certificate).

Round 2
Reviewer 1 Report
The revised manuscript is much improved. However, several parts do not make sense, where the reviewer is afraid that the authors seem not understand what the results of statistics mean.
1. Line 367 page 9
“miniCASCO (OR=0.9998; 95%CI: 0.9995 to 1.0001; p=0.0310)”.
If the 95% CI is across 1.0, the p-value should not be less than 0.05. The calculation should be incorrect.
2. Table 3
Most of ORs by miniCASCO subscale (i.e., anorexia, BWC, PHP, and QoL subscales) are higher than 1.0, which means that high scale (namely, higher likelihood of having cachexia) is associated with higher likelihood of having “disease control” (versus “progressive disease”). It means that the patients with cachexia tend to be in “disease control”. It seems not make sense. Furthermore, although 4 of 5 subscales have positive association with “disease control”, the total miniCASCO score, which is a sum of 5 subscales, have negative association with “disease control”, as stated by authors in the text. It is not possible unless the remaining subscale (namely, IMD subscale) has much more impact than the other 4 subscales, but it seems not the case because regression coefficient is not so different among subscales.
3. The reviewer is not sure that that “dataless abstract” (except the number of patients) is possible for publication. It depends on the editorial decision.
4. Figure A3 represents the box-whisker plot, which means nonnormal distributions, whereas figure A2 and A4 represent normal distributions. In this case, ANOVA is not indicated.
Author Response
Point-by-point reply to Reviewer 1 comments-ROUND 2
The revised manuscript is much improved. However, several parts do not make sense, where the reviewer is afraid that the authors seem not understand what the results of statistics mean.
- Line 367 page 9
“miniCASCO (OR=0.9998; 95%CI: 0.9995 to 1.0001; p=0.0310)”.
If the 95% CI is across 1.0, the p-value should not be less than 0.05. The calculation should be incorrect.
Reply: I agree with your punctual correction. I am sorry, there was an error in typing statistical results (OR and 95% CI). I have corrected the text (see lines 442-443 of the revised tracked changes version.
- Table 3
Most of ORs by miniCASCO subscale (i.e., anorexia, BWC, PHP, and QoL subscales) are higher than 1.0, which means that high scale (namely, higher likelihood of having cachexia) is associated with higher likelihood of having “disease control” (versus “progressive disease”). It means that the patients with cachexia tend to be in “disease control”. It seems not make sense. Furthermore, although 4 of 5 subscales have positive association with “disease control”, the total miniCASCO score, which is a sum of 5 subscales, have negative association with “disease control”, as stated by authors in the text. It is not possible unless the remaining subscale (namely, IMD subscale) has much more impact than the other 4 subscales, but it seems not the case because regression coefficient is not so different among subscales.
Reply: Thank you very much for your punctual corrections. I am very sorry, I have checked the results, and I found that in making the revision I have typed erroneously the results for BWC; PHP, and QoL, both in the text and tables. I have checked for typing since I have also noticed that for QoL I have reported a negative coefficient with an OR higher than 1, which has not sense. As regard ANO subscale, I confirm that I have obtained a positive regression coefficient with a OR higher than 1. As regard the calculation of the miniCASCO, I confirm it is the sum of the subscales, but, as reported in the miniCASCO validation paper and website cited in the methods section, the different subscales and the different questions have a different weight. I have added a sentence on this point in the Methods Section (paragraph 2.4.3): “The total score of miniCASCO is the sum of each single subscale, where each subscale contributes to the final score with a different weight as indicated in detail in the vali-dation paper [32].” See revised values at lines 431-444 and in the revised table 3 of the revised tracked changes version.
- The reviewer is not sure that “dataless abstract” (except the number of patients) is possible for publication. It depends on the editorial decision.
Reply: I have added the main results, including statistical detail, in the abstract. See lines 40-50 of the revised tracked changes version.
- Figure A3 represents the box-whisker plot, which means nonnormal distributions, whereas figure A2 and A4 represent normal distributions. In this case, ANOVA is not indicated.
Reply: Thank you for your observation. Since OS had a nonnormal distribution, I used Kruskall-Wallis test to compare it between categories. Accordingly, I revised the data both in the text and in the legend of the Figure A3 in the revised appendix.
